Earth System Science Data Discussions Open Access

# 1 Global soil moisture storage capacity at 0.5° resolution
# 2 for geoscientific modelling

Kang Xie[1,2,3], Pan Liu[1,2,3] *, Qian Xia[1,2,3], Xiao Li[1,2,3], Weibo Liu[1,2,3], Xiaojing
Zhang[1,2,3], Lei Cheng[1,2,3], Guoqing Wang[4], Jianyun Zhang[4]
[1]State Key Laboratory of Water Resources and Hydropower Engineering Science, Wuhan University,
Wuhan 430072, China.
[2]Hubei Provincial Key Lab of Water System Science for Sponge City Construction, Wuhan University,
Wuhan 430072, China.
[3]Research Institute for Water Security (RIWS), Wuhan University, Wuhan 430072, China.
[4]State Key Laboratory of Hydrology-Water Resources and Hydraulic Engineering, Nanjing Hydraulic
Research Institute, Nanjing 210029, China.
*Correspondence to*: Pan Liu (liupan@whu.edu.cn)
**Abstract.** Soil moisture storage capacity (SMSC) links the atmosphere and terrestrial ecosystems,
which is required as spatial parameters for geoscientific models. However, there are currently no
available common datasets of the SMSC on a global scale, especially for hydrological models since
conventional evapotranspiration-derived estimates cannot represent the extra storage capacity for the
lateral flow and runoff generation. Here, we produce a dataset of the SMSC parameter for global
hydrological models. Joint parameter calibration of three commonly used monthly water balance
models provides the labels for a deep residual network. The global SMSC is constructed based on the
deep residual network at 0.5° resolution by integrating 15 types of meteorological forcings, underlying
surface properties, and runoff data. SMSC products are validated with the spatial distribution against
root zone depth datasets and validated in the simulation efficiency on global grids and typical
catchments from different climatic regions. We provide the global SMSC parameter dataset as a
benchmark for geoscientific modelling by users.

## 25  1. Introduction

Soil moisture in the root zone layer is one of the vital hydrological variables in Earth system
dynamics (Wang-Erlandsson et al., 2022). Soil moisture storage capacity (SMSC[$L$]) is defined as the
total amount of water stored in the soil within the plant root zone, one of the essential parameters



linking the atmosphere and terrestrial ecosystems in the hydrological components (Chen, 2014;
Mccormick et al., 2021). The rooting depth of the plant cover determines the extent to which vegetation
returns water into the atmosphere via plant transpiration (Kleidon, 2004). A deeper SMSC means a
larger volume of water stored in the soil and, therefore, a larger reservoir of water available for crops to
draw from. Additionally, SMSC determines the storage and outflow capacity of water and is one of the
comprehensive parameters that affect the rainfall-runoff relationship. Therefore, the global
parameterization of SMSC is necessary for geoscientific modelling. The SMSC has been widely
applied in the hydrological models, such as Xinanjiang Model (Xie et al., 2020b; Zhao, 1992),
Dynamic Water Balance Model (DWBM) (Wang et al., 2011; Zhang et al., 2008), Snowbelt-based
Water Balance Model (SWBM) (Wang et al., 2014), and Time-variant Gain Model (TVGM) (Wang et
al., 2009; Xia et al., 1997), etc. These hydrological models at different spatial and temporal scales have
the same runoff generation structure, and SMSC becomes an essential parameter in the hydrological
process (Bai et al., 2015; Jaiswal et al., 2020).
Broadly, previous studies have investigated conventional approaches to estimating the spatial
distribution of the storage capacity in the root zone (Fan et al., 2017; Wang-Erlandsson et al., 2016;
Yang et al., 2016). However, there is currently no consensus on the estimation of SMSC. Even with
rooting depth measurements in situ from various field and laboratory observations, it is difficult to
estimate the root zone storage capacity due to uncertainty in root density, hydrological activity, and
horizontal spatial heterogeneity in soil data. The conventional calibration approach is only suitable for
applications at the catchment scale, and therefore challenges remain with parameter equifinality. The
conventional cumulative water deficit approach usually estimates soil plant-available water storage
capacity from remote-sensing-based precipitation and evapotranspiration fluxes (Stocker et al., 2021).
However, evapotranspiration-derived estimates of root-zone depth cannot represent the lateral flow and
runoff generation. Soil water is not only absorbed by vegetation from root soil and stems for
evaporation but also retains more capacity for runoff generation and groundwater flow. Overall, to our
knowledge, little attention has been paid to quantifying a common global SMSC parameter from the
perspective of the rainfall-runoff relationship in hydrological and land surface models (Beck et al.,
2015a; Beck et al., 2015b; Nijssen et al., 2001). Conventionally estimated SMSC datasets are difficult





to obtain the advantage of the model performance in global hydrological models.

Intense temporal unevenness and spatial heterogeneity have led to myriad problems in the

parameterization solution (Ming et al., 2017; Blschl et al., 2019). Most global hydrological models are
not calibrated or use prior knowledge to adjust SMSC parameters at large catchment scales since
calibrations become computationally intensive under large amounts of data and uncertainty from basin
characteristics (Wang et al., 2021). Essential parameters are even calibrated uniformly to subbasins
over the entire watershed or only against regional data. During the last decade, much work has been
done on the parameters and spatiotemporal boundaries of models (Chen, 2014; Imhoff et al., 2020;
Samaniego et al., 2010; Vinogradov et al., 2011). The results demonstrated that the spatial distributions
of regionalized parameters matched well with the climate and physiographic properties (Gentine et al.,
2012). Samaniego et al. (2010) proposed a multiscale parameter regionalization (MPR) technique by a
nonlinear transfer function and achieved the parameter transferability across the ungauged areas. Tsai et
al. (2021) proposed a differentiable parameter learning (DPL) framework that efficiently learns a global
mapping between dynamic inputs and hydrological parameters, and a deep learning model is trained to
generate the generic parameters. Hence, many approaches have demonstrated the necessity and the
feasibility of considering spatial heterogeneity in the hydrological process in quantifying a common
global SMSC parameter dataset.

This study seeks the global construction of the common SMSC parameter while accepting the

existing differences among hydrological models. The structure of the construction method is shown in
**Figure 1**. Specifically, the spatial distribution of SMSC parameters is obtained by the Shuffled
Complex Evolution (SCE-UA) algorithm for the joint calibration against an observation-based global
gridded runoff (GRUN) dataset. A deep residual network (ResNet) is used to learn the relationship
between the input factors and the regression SMSC parameters to consider spatial heterogeneity. The
results of the joint calibration provide the labels for the training of ResNet. Finally, the SMSC
parameter dataset is spatially constructed based on the pre-trained ResNet on the grid-scale to fill in
data empty areas where SMSC parameters are not available by the calibration approach. The global
runoff database center (GRDC) station streamflow data validates the global SMSC parameter dataset.
Solving the problem of common parameter datasets can help improve the simulation accuracy of global



hydrological models and help explore the physical meaning of model parameters associated with
surface heterogeneity. The global modeling community would benefit significantly from more common
parameter datasets.
[Please insert **Figure 1** here]
**2. Data**
Meteorological forcings and underlying surface properties affect the soil water storage capacity
from hydrological processes, soil structure, and plant root zone. The model inputs include 15 variables
such as global meteorological data, soil and vegetation data, topographical data, and streamflow
characteristics. **Table 1** provides the data sources used in the study.
[Please insert **Table 1** here]
There are two different types of inputs, a continuous value input represented by precipitation and
elevation and a categorical input represented by soil type and vegetation type. The model inputs are
standardized. Time series values of meteorological data are used as inputs of the hydrological model.
The multi-year averages of meteorological data are used as the spatial inputs to the deep learning
model. Monthly measurements cover the year from 1902 to 2014 in the global grids. The data for the
first year is used for warm-up, 80 years for calibration, and the remaining 30 years for validation.
**3. Methods**
**3.1 Gridded-based monthly water balance models**
Water balance models are one of the attractive models among the available hydrological
simulation techniques, offering flexibility and comprehensibility (Abdollahi et al., 2017;
Rodríguez-Huerta et al., 2020; Schaake et al., 1996). Water balance models can estimate daily, monthly,
and annual hydrological variables and processes by considering soil moisture. The advantages of
simple structure, fewer parameters, and fewer data requirements positively affect calibration and
regionalization.
Monthly water balance models simulate and predict the monthly runoff under different climatic
conditions (Do et al., 2020; Gui et al., 2019; Xiong et al., 2019). Monthly runoff processes differ from

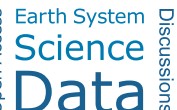

daily runoff because they generalize the stochastic uncertainty over a short time scale. Therefore, there
is no need to distinguish runoff yield and route in monthly water balance models, leading to simple
structures and straightforward applications (Zhang et al., 2018). Most monthly water balance models
have the concept of a water tank model (Bai et al., 2015; Singh and Woolhiser, 2002). This study
selects three monthly water balance models for the SMSC parameter.

(1) Dynamic Water Balance Model (DWBM)

The dynamic water balance model used in this study is the Budyko framework model by Wang et

al. (2011) and Zhang et al. (2008). The mean annual water balance can be modeled using the method of
Budyko (1958) by only considering dominant controls on evaporation. Fu (1981) developed the
following relationships for estimating mean annual evaporation:
$$\frac{E}{P} = 1 + \frac{E_0}{P} - \left[1 + \left(\frac{E_0}{P}\right)^\omega\right]^{1/\omega} \tag{1}$$
where $E$ is the mean annual actual evaporation, $E_0$ is the potential evaporation, and $\omega$ is a model
parameter with the range of $(1,\infty)$. The catchment is conceptualized as a system of two storages: root
zone storage and groundwater storage. Direct runoff can be calculated by rainfall $P(t)$ in time step $t$
deducting catchment rainfall retention $X(t)$
$$Q_d(t) = P(t) - P(t)F\left(\frac{X_0(t)}{P(t)}, \alpha_1\right) \tag{2}$$
where $F()$ is Fu's curve - Eq. (1), $\alpha_1$ is retention efficiency, i.e., a larger $\alpha_1$ the value will result in
more rainfall retention and less direct runoff. Evaporation E(t) can be calculated as
$$E(t) = W(t)F\left(\frac{E_0(t)}{W(t)}, \alpha_2\right) \tag{3}$$
where $W(t)$ is water availability, and $\alpha_2$ is a model parameter representing evaporation efficiency.
The soil water storage can now be calculated as:
$$S(t) = W(t)F\left(\frac{E_0(t)+SMSC}{W(t)}, \alpha_2\right) - E(t) \tag{4}$$
where $SMSC$ is the soil moisture storage capacity. Finally, the soil water storage is treated as a linear
reservoir so that the groundwater balance and baseflow can be modeled as:
$$Q_g(t) = K_g S(t-1) \tag{5}$$



$$G(t) = (1 - K_g)S(t-1) + R(t) \tag{6}$$
where $S(t)$ is groundwater storage, and $K_g$ is a constant model parameter.
(2) Snowbelt-based Water Balance Model (SWBM)
The snowbelt-based water balance model used in this study is the Yellow river water balance
model by Wang et al. (2014). According to the influencing factors of surface runoff yield, the
calculation formula of surface runoff is put forward by generalizing the two runoff generation
mechanisms:
$$Q_d(t) = K_s \frac{S(t)}{SMSC} P(t) \tag{7}$$
where $Q_d(t)$ is the direct surface runoff, $S(t)$ is the soil moisture, and $SMSC$ is the maximum soil
moisture storage capacity, and $K_s$ is the coefficient of surface runoff. It is assumed that the
underground runoff is a linear reservoir discharge. The underground runoff is calculated as follows:
$$Q_g(t) = K_g S(t-1) \tag{8}$$
where $Q_g(t)$ is the underground runoff, and $K_g$ is the coefficient of the underground runoff. The
evaporation capacity of the basin is equal to that of the water surface. The calculation of long-term
evaporation of the basin is based on the calculation model of soil evaporation as follows:
$$E(t) = E_m \frac{S(t-1)}{SMSC} \tag{9}$$
where $E(t)$ is the actual evaporation, $E_m$ denotes the evaporation capacity of the basin and is
calculated according to the meteorological data.
(3) Time-variant Gain Model (TVGM)
The relationship between rainfall and runoff is nonlinear. To grasp its nonlinear nature from
system theory, Xia et al. (1997) and Wang et al. (2009) proposed the time-variant gain model (TVGM)
model. The TVGM model can describe the nonlinear relationship between input and output of the
hydrological cycle system by introducing a time-varying gain factor. The direct surface runoff
generated by the catchment can be expressed as:
$$Q_d(t) = g_1 \left( \frac{S(t)}{SMSC} \right)^{g_2} P(t) \tag{10}$$



where  $S(t)$  is the soil humidity at the beginning of the period,  $SC$  is the saturated soil humidity,  $P$  is
rainfall,  $g_1$  and  $g_2$  are the related parameters of the time-varying gain factor, where  $g_1$  is the runoff
coefficient after soil saturation,  $g_2$  is the soil moisture influence coefficient. Soil moisture flow is
calculated as follows:
$$Q_g(t) = K_r[S(t-1) + S(t)]/2 \tag{11}$$
where  $K_r$  is the coefficient of soil moisture outflow. The actual evaporation is based on the
rainfall-evaporation model considering soil moisture as follows:
$$E(t) = PET(t)\left(\frac{S(t)}{SMSC}\right)^\gamma \tag{12}$$
where  $\gamma$  is the weight coefficient of evaporation.
**3.2 Parameter calibration strategy**

In principle, parameters in the hydrological model are constructed based on the interpretation of

the measured response in the catchment. However, for those parameters for which no measured values
are available, the initial values of the parameters can first be determined empirically or by referring to
previous results. Then the parameters are optimized according to the specific objectives against
simulation results. Processes at different scales interact and influence each other, leading to the
complexity of parameter calibration. The calibration will result in the spatially discontinuous parameter
in each basin. The calibration aims to consider the spatial interactions of the parameters but often
pursues the simulation accuracy too much since inputs are homogenized across catchments. Different
areas make it difficult for the parameters to converge to spatially continuous values. Therefore, the
parameters are calibrated on a spatial grid of the same area in this study. Research has shown that
calibration on the global grids can significantly reduce parameter discontinuities compared to
calibration on individual catchments (Xie et al., 2020a). The conceptual parameters in three monthly
water balance models (**Table 2**) are calibrated against the agreement between simulated and observed
hydrographs until the optimal value is obtained.

[Please insert **Table 2** here]

Two parameter calibration strategies are listed below, and the joint calibration strategy is

considered in this study.
**Individual calibration strategy**: Each model is calibrated separately across global grids. The
purpose is to find the similarities and differences in the SMSC parameter distribution of three different
model structures. The gridded runoff depth data is used as observations for the calibration. The gridded
global monthly runoff time series are obtained from the GRUN dataset on a 0.5 degrees grid covering
1902 to 2014 (Ghiggi et al., 2019a; Ghiggi et al., 2019b; Ibarra et al., 2020). The parameters calibrated
in the catchment are used as the initial values on catchment grids.
**Joint calibration strategy**: This procedure will calibrate all parameters of three models in a joint
calibration, and the SMSC parameters in each model are equal. The physical meaning of the parameters
can only be expressed in terms of the same values. There should be a value between the optimal values
of multiple models. This value has a physical meaning in terms of spatial continuity and can be
commonly considered for each model.
The SCE-UA algorithm, one of the common global optimization methods, is used for the
parameter calibration of monthly water balance models (Duan et al., 1994). The objective function is
selected as the least-squares method, i.e., Mean Square Error (MSE). The Kling-Gupta Efficiency
(KGE) is used to quantify the performance of the model simulations, which is a model evaluation
criterion that can be decomposed into the contribution of mean, variance, and correlation to model
performance (Gupta et al., 2009). KGE is calculated as follows:
$$\text{KGE} = 1 - \sqrt{(r-1)^2 + (\mu_{sim}/\mu_{obs} - 1)^2 + (\sigma_{sim}/\sigma_{obs} - 1)^2} \qquad (13)$$
where $r$ is the Pearson Correlation Coefficient, $\mu$ and $\sigma$ are the mean and common deviation of the
variables. KGE value ranges from $-\infty$ to 1, with a value closer to 1 indicating a better simulation
performance.
**3.3 Deep residual network**
Although the parameters obtained by the calibration at grid scale are accurate and have good
spatial continuity, it is still challenging to obtain parameters on many unsuitable grids due to the
limitations of the hydrological model in the ungauged area. It remains a daunting challenge to mine the
hidden information from a large amount of data because of the inherent physical variability in complex
physical mechanisms (Clark et al., 2016; Zhang and Liu, 2021). Driven by the increasingly powerful
performance of computers and big data, statistical and non-inferential deep learning methods enable
machines to have the same ability to analyze and learn as human beings (Kadow et al., 2020; Karpatne
et al., 2018; Sit et al., 2020). Recent case studies have revealed that deep learning networks have
succeeded in geoscience fields (Karpatne et al., 2018; Xie et al., 2021). It has been widely used for
spatial missing data (Kadow et al., 2020), spatial downscaling (Jiang et al., 2021; Nearing et al., 2021),
rainfall simulation improvement (Liu et al., 2020), and spatial phenomena prediction (Pan et al., 2019).
Convolutional neural networks (CNNs) can automatically learn features from massive data and
generalize the results to unknown domains of the same type (Shin et al., 2016). The convolution and
pooling layers in CNNs only work on a local neighborhood, which helps to capture local geometric
features and spatial patterns and extract larger-scale representations in deeper layers (Shen, 2018). The
filters are shared when calculating the neurons of the same depth slice, which reduces the number of
parameters and makes them easier to train.
A deep residual network, one of the specific types of CNN method, can automatically learn
features from large-scale data and generalize the results to anonymous data of the same type (He et al.,
2016). However, CNN has reached saturation accuracy when the number of layers deepens, called
degradation. The network's performance deteriorates, and it is challenging to train shallow networks by
backpropagation because the gradient dissipation is more severe. ResNet solves this problem by
making it easier for gradients to flow into external networks. The structure of ResNet adds the residual
mapping and the identity mapping through shortcut connections. If the network has reached the optimal
level and continues to deepen, the residual mapping will be pushed to zero, leaving only identity
mapping. Theoretically, the network has been in the optimal state, and the network performance does
not decrease with increasing depth. Finally, the gradient vanishing can be avoided, and the network can
be deepened. ResNet provides a new approach to learning SMSC parameters using more information
from similar grids (Zhuo and Tan, 2021).
CNN local connection means that each neuron is connected to only one region of the input neuron,
and the filters used by CNN to compute neurons of the same depth slice are shared. These



characters are similar to the hydrological parameters and the spatial characteristics of the input data.
From the conventional statistic method to the deep paradigm, ResNet has the following three
outstanding advantages against conventional statistic methods.
(1) ResNet is provided with the more vital generalization ability. Conventional statistic methods
cannot explore the complex inner connections of the soil water process, while ResNet avoids directly
interpreting the physical meaning of the parameters firstly.
(2) More input variables are used in ResNet. Conventional root depth calculations use only
precipitation and evaporation, while both meteorological forcings, underlying surface properties, and
runoff data are considered in ResNet.
(3) ResNet has faster speed and higher performance. Conventional statistic methods cannot learn
complex interactions and are slow to compute. However, parallel computing is used in ResNet, and the
network is complex but much faster. The model is run on a GPU (Nvidia Tesla V100 16GB) cluster and
takes 758 microseconds per step, about one hour on all global 0.5-degree grids.
**3.4 Training and testing**
The SMSC parameters on the global grids obtained by the calibration algorithm are taken as the
target labels of the model. On grids with KGE greater than 0, SMSC parameters can be obtained by
calibrating the hydrological model. However, the hydrological model cannot be built in some areas
where the model is not applicable, such as highly arid areas. Areas with KGE less than 0 are masked.
On grids with KGE greater than zero, the samples are divided into the training set and test set
according to the ratio of 7:3.
**Table 3** shows the learning performance of the training and test sets for different image windows.
The results show that the recognition network is poor if the image window is too small. The effect of
$10 \times 10$ image windows is better than that of $5 \times 5$ grid windows, and the effect of 100 surrounding
grids on the center grid can be considered for $10 \times 10$ windows. The Correlation coefficient ($R^2$) of the
test set increases from 0.59 to 0.76. The computational burden from the increase in image windows is
no longer as cost-effective as the increase in inefficiency.
[Please insert **Table 3** here]





**3.5 Permutation importance**
The deep learning network is often considered a black-box model, and here interpretation
techniques are used in order to better understand the underlying relationships tapped by deep learning.
Permutation feature importance is a model inspection technique widely used for deep learning
networks (Altmann et al., 2010). For a fitted predictive model, permutation importance can compute
the reference score of the model on the dataset. The importance $i_j$ for each feature $f_j$ defined as:
$$i_j = s - \frac{1}{K}\sum_{k=1}^{K} s_{k,j} \tag{14}$$
where $s$ is the reference score of the model, for instance, the accuracy for a classifier or the correlation
coefficient ($R^2$) for a regressor, $k$ is each repetition in input factors.
**4. Evaluation of the global soil moisture storage capacity**
**4.1 Comparison of the spatial distribution with other parameter datasets**
**Figure 2**a shows the SMSC values jointly calibrated by setting the SMSC parameters of the three
models to be the same. The results show that combined objectives for the calibration of three models
are relatively stricter, with only 45% of the grid KGE greater than zero, which is called the labeling
area (the opposite corresponding to the constructing area). The SMSC parameters are larger in humid
areas and smaller in arid areas. The hydrological model is no longer applicable outside the labeling area,
such as semi-arid and cold regions. **Figure 2**c shows the probability density distribution of SMSC
parameters calibrated in the labeling area. It can be found that the distribution of the jointly calibrated
SMSC parameters is consistent with the distribution of the individually calibrated SMSC parameters.
**Figure 2**b shows the spatial distribution of the global SMSC parameters both in the labeling area and
the constructing area. The constructed SMSC is also larger in humid regions and smaller in arid regions.
The parameters are larger in high-altitude regions. **Figure 2**d and **Figure 2**f show the variation of
global SMSC with latitude and longitude. The results show that the global SMSC is largest at the
equator and decreases toward the poles. **Figure 2**e shows the probability density function of the global
SMSC with a double-peak distribution. The first peak corresponds to the arid region, and the second
peak corresponds to the humid region.



[Please insert **Figure 2** here]

We compared the spatial distribution of global SMSC with other parameter datasets. **Figure 3**

shows estimations of global root zone parameters from previous studies and compares them to global
SMSC. Root zone storage capacity at 0.5° resolution (SR_CRU$_{Wang-Erlandsson}$, **Figure 3**a) is estimated by
computing the maximum moisture deficit with independent energy balance equations by satellite-based
evaporation from Wang-Erlandsson et al. (2016). Rooting depth at 1.0° resolution (SR$_{Schenk}$, **Figure 3**c)
estimates the rooting depth that contains 95% of all roots from Schenk et al. (2009). By comparing
these datasets with SMSC, we can see both agreements and significant differences. The purpose is the
geographic comparison to other soil moisture storage capacity estimates. These comparisons are
expected to find differences in the spatial distribution between root depth and soil moisture storage
capacity. All datasets show a relatively similar spatial distribution, decreasing from the equator to the
poles. All datasets have smaller values in the tropical rainforest region near the equator than our SMSC
product. SR$_{Schenk}$ tends to overestimate SMSC parameters in the Sahara Desert, Arabian Desert, and
Western Australian Desert. ET-derived estimated water storage capacity might be relatively small in
some areas. **Figure 3**b shows that the roots of vegetation (SR_CRU$_{Wang-Erlandsson}$) may not be so deep,
especially in the humid region of the equatorial, but our proposed SMSC data is deeper in the humid
region. The findings indicate that ET-derived estimates of root-zone depth are unable to represent the
lateral flow and runoff generation. Soil water is not only absorbed by vegetation from root soil and
stems for evaporation but also retains more capacity for runoff generation and groundwater flow.

[Please insert **Figure 3** here]

**4.2 Model performance of runoff depth in global grids**

We tested runoff depth simulations of the global SMSC dataset on global grids. Gridded-based

monthly water balance models are established on each 0.5 ° and 0.5 ° grid over the global terrestrial
land. The SMSC parameters in these models adopt the proposed global SMSC dataset, while other
parameters are recalibrated. The model parameter SMSC among the three monthly water balance
models is the same in the proposed dataset constructed by CCN. This parameter is no longer
recalibrated in further modeling. The inputs are monthly precipitation and evaporation for each grid.
Monthly runoff depths of GRUN on this grid are used as the observations for model evaluation. **Figure**





**4**a-c presents the distribution of simulation accuracy for three water balance models on the global grids. The models perform well in the humid region, semi-humid region, and most of the semi-arid region. The KGE performance of the models is significantly better in the humid region than in the semi-humid region and most of the semi-arid region. **Figure 4**d shows the KGE probability density distributions of three models. The results demonstrate that the TVGM model performs the best, with 20% of the grids having KGE values above 0.80 and 40% above 0.60. The results also indicate two distinct peaks in the KGE distribution of the SWBM model. The peak on the left represents the poor KGE of the SWBM model in the semi-humid and most semi-arid regions. **Figure 4**e shows the cumulative probability density distribution of KGE for three models on global grids, and **Figure 4**f shows the KGE box. What stands out in the figure is that the TVGM model has the best KGE, where the average KGE can reach 0.55, while the SWBM model is the worst.

As shown in **Figure 4**, the results indicate that TVGM and DWBM models perform better in the cold region. These three models do not take the temperature as the input, and therefore the snowbelt module is not considered. All three models do not perform very well in arid and semi-arid areas. The water balance model is challenging to simulate monthly runoff in arid areas because of the mismatch of the rainfall-runoff relationship.

[Please insert **Figure 4** here]

**4.3 Model performance of streamflow in typical catchments**

Station streamflow is used for the validation of global SMSC parameters. The GRDC dataset is a unique collection of river discharge data on a global scale (Peel et al., 2004; Peel et al., 2001). It contains daily and monthly river discharge data from over 10,000 stations worldwide. The selected validation basins require a basin area of more than 10,000km$^2$ and a monthly runoff record of more than five years from 1991 to 2010. Finally, data from 20 stations in different climatic regions are selected for validation. These 20 significant rivers are distributed in five different climate zones. **Table 4** lists the simulated KGE of three models in 20 typical catchments, and the average simulation accuracy is more than 0.65. **Figure 5** shows the dots of simulated streamflow versus observed streamflow during the validation periods.

[Please insert **Table 4** here]



[Please insert **Figure 5** here]

The spatial patterns presented by the three models would be extraordinarily different if the three

models were directly applied in the catchment according to the lumped model since different catchment

areas influence them. Although this approach achieves good simulation accuracy, it does not consider

the physical significance and spatially seamless alignment. However, the constructed global SMSC

parameters have an excellent spatial continuity. The average values of the constructed SMSC parameter

are calculated in 20 basins in different climatic regions as the recommended value of the parameters.

**Figure 6** shows the simulation accuracy of SR_CRU$_{Wang-Erlandsson}$ parameters in 20 basins compared

with the SMSC parameters constructed in this study. The results show higher KGE performance of the

constructed SMSC parameters in the three selected monthly water balance models in the 20 selected

basins. Labels of the SMSC parameters are derived from the results of the model parameters, and more

input information is considered in the construction. The purpose of the comparison is to evaluate the

proposed dataset from the perspective of hydrological models. SMSC estimated from the model's

perspective has achieved higher KGE performance and is more practical. The CRU$_{Wang\_Erlandsson}$ dataset

is estimated using only two data types, precipitation, and evaporation, but it lacks model validation.

Even if actual evaporation is also used in the calculation, the SMSC calculated by this method may not

be able to simulate evaporation accurately because it lacks a model basis. On the contrary, our product

utilizes a hydrological model, which can simultaneously simulate evaporation, runoff, and soil water

content and achieve water balance.

[Please insert **Figure 6** here]

**4.4 The sensitivity of input factors selection**

Model input factors of the deep residual network include 15 variables affecting SMSC such as

global meteorological data, soil and vegetation data, topographical data, and streamflow characteristics.

These available factors, including meteorological forcings and underlying surface properties.

Meteorology data include precipitation, potential evaporation, and near-surface temperature, which

influence these processes such as evaporation, transpiration, and runoff in the water cycle. Soil data

include soil thickness, root zone depth, soil type, and types of land use, which influence the soil

structure. As the permutation importance estimate of **Figure 7** below shows, the most significant



factors influencing the spatial construction of water storage capacity parameters in global grids are the
precipitation and the type of land use. The precipitation, as the dominant factor in the spatial evolution
of the SMSC parameter, explains more than 60% of the spatial distribution of SMSC. The precipitation
and the type of land use directly influence the root zone depth and porosity of vegetation in different
areas.

[Please insert **Figure 7** here]

**5. Uncertainty of the data**

However, there exists some uncertainty to this dataset. Meteorological data mainly include

precipitation, temperature, and evaporation. Hydrological models are very sensitive to meteorological
data, especially precipitation data. Firstly, data have intensively spatial and temporal variability. Most
of the gird-based meteorological product comes from scattered observation sites, which cannot fully
describe the spatial characteristics of features. Especially for large watersheds, the observation stations
in the watershed cannot well represent the spatiotemporal changes, which may eventually affect the
results of the model simulation. Secondly, there are also errors in the measurement data of the
observation station, which leads to the uncertainty of the input data.

In addition to meteorological data as input data, spatial data such as digital elevation, land use data,

soil type data, etc. are usually required. The accuracy of the spatial data to describe the characteristics
of the watershed is the premise of the accurate simulation of the model. The resolution of the elevation,
the accuracy of the land use type and the accuracy of the soil data type all have a certain impact on the
simulation results. The level of resolution affects the extraction of parameters of the study watershed
characteristics (slope, slope aspect, water and sediment migration direction, confluence network,
watershed boundary, etc.), and ultimately affects the accuracy of the product.

The nonlinearity of the model structure and the correlation of parameters make the model solution

space possible to have multiple local optimal solutions. The above effects all lead to large uncertainties
in the process of watershed runoff simulation in the distributed hydrological model.



**6. Recommendations and limitations for the use of the data**

This product is only limited by the current climatic conditions and ignores future changes. We estimated the SMSC based on the meteorological forcings, underlying surface properties, and runoff dataset over the calibration and validation period 1902-2014. Results may change when using data from different periods. Recent studies show that soil water storage capacity in the root zone changes with climate change and deforestation. The vegetation changes the ability to utilize subsoil moisture storage and tree cover to respond to arid climates. Additionally, the proposed dataset provides the global SMSC parameter dataset mainly for the water balance models at a monthly scale. At the current stage, it does not provide insights on quality simulations of low flow and high flow on a daily or hourly scale.

**7. Code and data availability**

A global terrestrial SMSC dataset with 0.5° spatial resolution is now available. The global construction map of SMSC in this study can be gathered from an open-access data server. All input factors and the global SMSC data are publicly available as NetCDF files or downloaded from smsc_data.zip at Zenodo (https://doi.org/10.5281/zenodo.5598405, Xie (2021)). Python codes are available to calculate the basin average SMSC value from grid values in any interested basin on a global scale. The Fortran codes for the parameter calibration of gridded-based global monthly water balance models are available at https://github.com/xiekangwhu/SMSC_monthly_water_balance_models. The Python codes of the deep residual network we developed for the global construction map of SMSC are available at https://github.com/xiekangwhu/SMSC_deep_residual_network.

**8. Conclusions**

In this paper, a new global SMSC dataset for global hydrological models is constructed by the deep residual network at 0.5° resolution by integrating 15 types of meteorological forcings, underlying surface properties, and runoff data. Compared with SR_CRU$_{Wang-Erlandsson}$ and SR$_{Schenk}$ dataset, the results show that the accuracy of the three gridded-based monthly water balance models from high to



low is the proposed SMSC, SR_CRU$_{Wang\text{-}Erlandsson}$ and SR$_{Schenk}$. Through the interpretation technique of
the deep residual network, the most significant factors influencing water storage capacity parameters in
global grids are precipitation and land use.

**Author contribution**

Kang Xie performed writing - original draft, conceptualization, methodology, and software.
Pan Liu performed writing - review and editing, conceptualization, and supervision.
Qian Xia performed the methodology of hydrological models.
Xiao Li performed data processing and coding.
Weibo Liu performed writing - review and editing.
Xiaojing Zhang performed the methodology of the parameter calibration strategy.
Lei Cheng performed writing - review and editing.
Guoqing Wang performed writing - review and editing.
Jianyun Zhang performed writing - review and editing, and supervision.

**Competing interests**

The authors declare that they have no competing interests.

**Disclaimer**

Publisher's note: Copernicus Publications remains neutral with regard to jurisdictional claims in
published maps and institutional affiliations.

**Acknowledgments**

The authors appreciate the help from the Supercomputing Center of Wuhan University for
providing the necessary guides to perform the numerical calculations of this study on the
supercomputing system.



**Financial support**
This study was supported by the Joint Funds of the National Natural Science Foundation of China
(Grant No. U1865201), National Natural Science Foundation of China (Grant No. 51861125102),
Innovation Team in Key Field of the Ministry of Science and Technology (Grant No. 2018RA4014),
and the Young Scientists Fund of the National Natural Science Foundation of China (Grant No.

52109030).

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



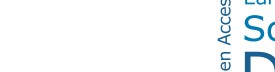



**Table 1. Research data and sources.**

| Data type | Data | Spatial resolution | Time span | Data/product sources |
|---|---|---|---|---|
| Meteorology | Precipitation Potential evaporation Near-surface temperature | 0.5 degree | January 1901 - December 2018 (monthly) | Cru TS 4.03 monthly high-resolution grid data https://data.ceda.ac.uk/badc/cru/data/cru_ts/cru_ts_4.03 |
| Soil and vegetation | Soil thickness | 0.5 degree | / | Global 1km grid soil, weathering layer, and sedimentary layer thickness published by ORNL in 2016 https://daac.ornl.gov/SOILS/guides/Global_Soil_Regolith_Sediment.html |
| | Root zone depth | | | Global root zone depth products released by Stockholm university in 2016 http://dx.doi.org/10.5194/hess-20-1459-2016-supplement |
| | Soil type | | | Upper and lower global soil type data released by USDA https://www.nrcs.usda.gov/wps/portal/nrcs/detail/soils/use/ |
| | Types of land use | | | AVHRR land-use types issued by NOAA |
| Topography | Slope Altitude Composite terrain index CTI | 0.5 degree | / | GMT global 0.5-degree terrain data released by ISLSCP in 2010 https://daac.ornl.gov/cgi-bin/dsviewer.pl?ds_id=1007 |
| Runoff characteristic | Average flow Runoff coefficient Baseflow coefficient 1% flood discharge | 0.5 degree | / | GSCD global runoff data set released by GloH2O in 2015 http://www.gloh2o.org/gscd/ |
| Runoff | Runoff of catchment stations | / | January 1991 - December 2010 (monthly) | GRDC global runoff data center (Including data from more than 10000 stations around the world) https://www.bafg.de/GRDC/EN/Home/homepage_node.html |
| | Grid runoff | 0.5 degree | January 1902 - December 2014 (monthly) | GRUN global grid runoff depth database released by the Federal Institute of technology https://doi.org/10.6084/m9.figshare.9228176 |





**Table 2. Conceptual parameters to be calibrated in hydrological models.**

| Model | Parameter | Physical meaning | Minimum boundary | Maximum boundary |
|---|---|---|---|---|
| DWBM | $SMSC$ | Soil moisture storage capacity (mm) | 0 | 1000 |
| | $\alpha_1$ | Retardation coefficient | 1 | 5 |
| | $\alpha_2$ | Evaporation coefficient | 1 | 5 |
| | $K_g$ | Underground runoff coefficient | 0.01 | 1 |
| SWBM | $SMSC$ | Soil moisture storage capacity (mm) | 0 | 1000 |
| | $K_s$ | Surface runoff coefficient | 0.1 | 1 |
| | $K_g$ | Underground runoff coefficient | 0.01 | 1 |
| TVGM | $SMSC$ | Soil moisture storage capacity (mm) | 0 | 1000 |
| | $g_1$ | Runoff coefficient after soil saturation | 0.02 | 1.0 |
| | $g_2$ | Soil moisture influence coefficient | 1.0 | 5 |
| | $K_r$ | Soil moisture outflow coefficient | 0.005 | 1 |
| | $\gamma$ | Evaporation conversion index | 0.1 | 1 |




**Table 3. Results of construction model of global soil moisture storage capacity (SMSC) parameters.**

| Image window | Time interval | Loss function | Evaluating indicator | |
| --- | --- | --- | --- | --- |
| | | $MSE$ | $MAE$ | $R^2$ |
| 5×5 | Training set | 0.0032 | 0.0411 | 0.8932 |
| | Test set | 0.0170 | 0.0666 | 0.5935 |
| 10×10 | Training set | 0.0021 | 0.0341 | 0.9139 |
| | Test set | 0.0061 | 0.0510 | 0.7597 |







**Table 4. Validation of global SMSC parameters in typical catchments.**

| Number | Site name | Longitude | Latitude | Drainage area (km²) | River | KGE (%) of DWBM model | | KGE (%) of SWBM model | | KGE (%) of TVGM model | | Basin average SMSC (mm) |
|---|---|---|---|---|---|---|---|---|---|---|---|---|
| | | | | | | Cal[1] | Val[2] | Cal | Val | Cal | Val | |
| 1196551 | Beibrug | 29.99 | −22.23 | 201001 | Limpopo | 47.06 | 74.15 | 53.25 | 69.67 | 43.11 | 41.67 | 149.84 |
| 2181500 | Zhimenda | 96.6 | 33.43 | 137704 | Tongtian | 49.54 | 72.26 | 69.95 | 77.82 | 84.71 | 80.09 | 121.48 |
| 2181900 | Datong | 117.62 | 30.77 | 1705383 | Yangtze | 55.87 | 84.52 | 86.48 | 91.88 | 85.75 | 91.16 | 206.85 |
| 2260500 | Sagaing | 96.1 | 21.98 | 117900 | Irrawaddy | 78.27 | 76.35 | 70.93 | 68.64 | 63.22 | 56.99 | 365.65 |
| 2694450 | Waegwan | 128.39 | 36 | 11195 | Naktong | 67.81 | 58.89 | 66.63 | 41.78 | 77.99 | 44.93 | 231.37 |
| 3268270 | Caimancito | −64.47 | −23.73 | 25800 | San Francisco | 67.54 | 78.11 | 53.66 | 83.31 | 63.44 | 63.55 | 228.89 |
| 3618090 | Cucui | −66.85 | 1.22 | 61781 | Negro | 69.13 | 72.83 | 69.54 | 67.53 | 89.36 | 89.72 | 226.27 |
| 3624120 | Gaviao | −66.85 | −4.84 | 162000 | Jurua | 49.13 | 66.07 | 71.06 | 69.88 | 88.35 | 80.24 | 532.84 |
| 3627030 | Manicore | −61.30 | −5.82 | 1126700 | Madeira | 87.15 | 71.24 | 68.83 | 72.46 | 73.24 | 86.55 | 370.19 |
| 3629000 | Obidos-Porto | −55.51 | −1.95 | 4640300 | Amazonas | 73.55 | 80.47 | 58.66 | 58.92 | 57.02 | 54.63 | 388.80 |
| 3629150 | Fortaleza | −57.64 | −6.05 | 358657 | Tapajos | 39.03 | 49.10 | 87.55 | 74.90 | 75.24 | 63.62 | 428.36 |
| 3650745 | Ico | −38.87 | −6.41 | 12000 | Salgado | 39.22 | 46.87 | 54.93 | 63.24 | 58.56 | 94.82 | 392.60 |
| 4103800 | Eagle AK | −141.20 | 64.79 | 293965 | Yukon | 70.56 | 77.55 | 36.05 | 46.80 | 37.01 | 38.99 | 95.21 |
| 4115100 | Salem, OR | −123.04 | 44.94 | 18855 | Willamette | 86.86 | 89.73 | 80.01 | 86.28 | 59.46 | 66.52 | 475.76 |
| 4115201 | Beaver, OR | −123.18 | 46.18 | 665371 | Columbia | 58.60 | 47.52 | 79.14 | 76.35 | 88.81 | 74.00 | 358.43 |
| 4119100 | Paul, MN | −93.11 | 44.93 | 95312 | Mississippi | 22.95 | 14.15 | 60.29 | 23.76 | 60.88 | 55.06 | 186.30 |
| 4146281 | Verona, CA | −121.60 | 38.77 | 55040 | Sacramento | 43.65 | 64.24 | 70.63 | 60.40 | 89.41 | 88.84 | 344.22 |
| 5109170 | Rockfields | 142.88 | −18.20 | 10987 | Gilbert | 52.02 | 76.95 | 13.20 | 50.34 | 73.34 | 52.15 | 245.60 |
| 6335180 | Worms | 8.38 | 49.64 | 68827 | Rhine | 73.97 | 76.66 | 78.43 | 84.00 | 76.88 | 78.37 | 296.07 |
| 6342800 | Hofkirchen | 13.12 | 48.68 | 47496 | Danube | 56.53 | 46.58 | 61.31 | 53.67 | 69.49 | 61.30 | 247.41 |
| | Mean KGE | | | | | 59.42 | 66.21 | 64.53 | 66.08 | 70.76 | 68.16 | —— |

[1] Calibration period
[2] Validation period



**List of figures**

Figure 1. Structure of depth residual image recognition in convolutional neural network (CNN) model. (a) The spatial distribution of soil moisture storage capacity (SMSC) parameters is obtained by the Shuffled Complex Evolution (SCE-UA) algorithm for the joint calibration. (b) the relationship between the input factors and the regression SMSC parameters is learned by a deep residual network (ResNet). (c) the SMSC parameter dataset is spatially constructed based on the pre-trained ResNet on the grid-scale to fill in data empty areas.

Figure 2. Spatial distribution of labels and construction results for global soil moisture storage capacity (SMSC) parameters. (a) The spatial distribution of the label for deep learning of jointly calibrated SMSC values. (b) The spatial distribution of the constructed global SMSC parameters. (c) The probability density distribution of SMSC parameters calibrated in the labeling area. (e) The probability density distribution of constructed global SMSC parameters. (d) and (f) The distribution of variations of global SMSC with latitude and longitude.

Figure 3. Spatial distribution of other parameter datasets and the differences with global soil moisture storage capacity (SMSC) parameters. (a) The spatial distribution of root zone storage capacity at 0.5° resolution by Wang-Erlandsson et al. (2016). (c) The spatial distribution of rooting depth at 0.5° resolution by Schenk et al. (2009). (b) and (d) The differences between global SMSC parameters and other parameter datasets.

Figure 4. Global distribution of Kling-Gupta efficiency (KGE) simulated by GRUN runoff depth on grid-scale for three monthly water balance models. (a)The KGE of Dynamic Water Balance Model (DWBM). (b) The KGE of Snowbelt-based Water Balance Model (SWBM). (c) The KGE of Time-variant Gain Model (TVGM). (d) The probability density distributions of KGE for three models. (e) The cumulative density distributions of KGE for three models. (f) The boxplot of KGE for three models.

Figure 5. Monthly observed streamflow versus simulated streamflow for three monthly water balance models in typical catchments. Blue dots represent Dynamic Water Balance Model (DWBM). Green dots represent Snowbelt-based Water Balance Model (SWBM). Orange dots represent Time-variant Gain Model (TVGM).

Figure 6. Comparison of constructed SMSC parameter and root zone depth in runoff simulation for three monthly water balance models. The left figure represents the simulation accuracy during recalibration period. The right figure represents the simulation accuracy during validation period. The dots mean the average





values, and the lines mean the median values.
Figure 7. Spatial correlation between SMSC and input variables by the permutation importance
measure. Weights of permutation importance estimate represent the spatial correlation. The larger the
absolute value means the more relevant variables.

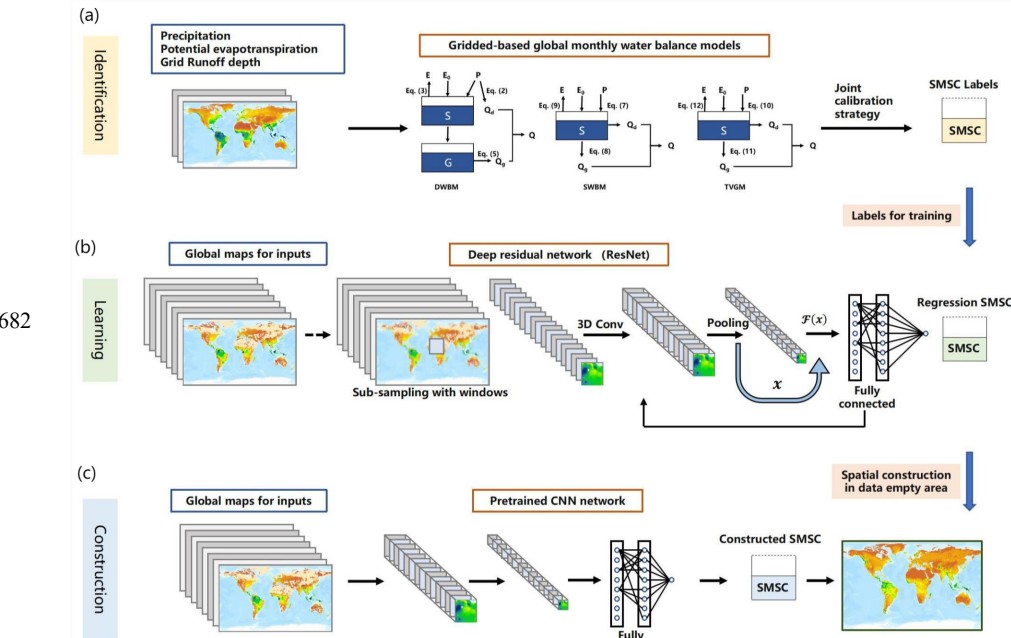

**Figure 1. Structure of depth residual image recognition in convolutional neural network (CNN) model. (a) The spatial distribution of soil moisture storage capacity (SMSC) parameters is obtained by the Shuffled Complex Evolution (SCE-UA) algorithm for the joint calibration. (b) the relationship between the input factors and the regression SMSC parameters is learned by a deep residual network (ResNet). (c) the SMSC parameter dataset is spatially constructed based on the pre-trained ResNet on the grid-scale to fill in data empty areas.**

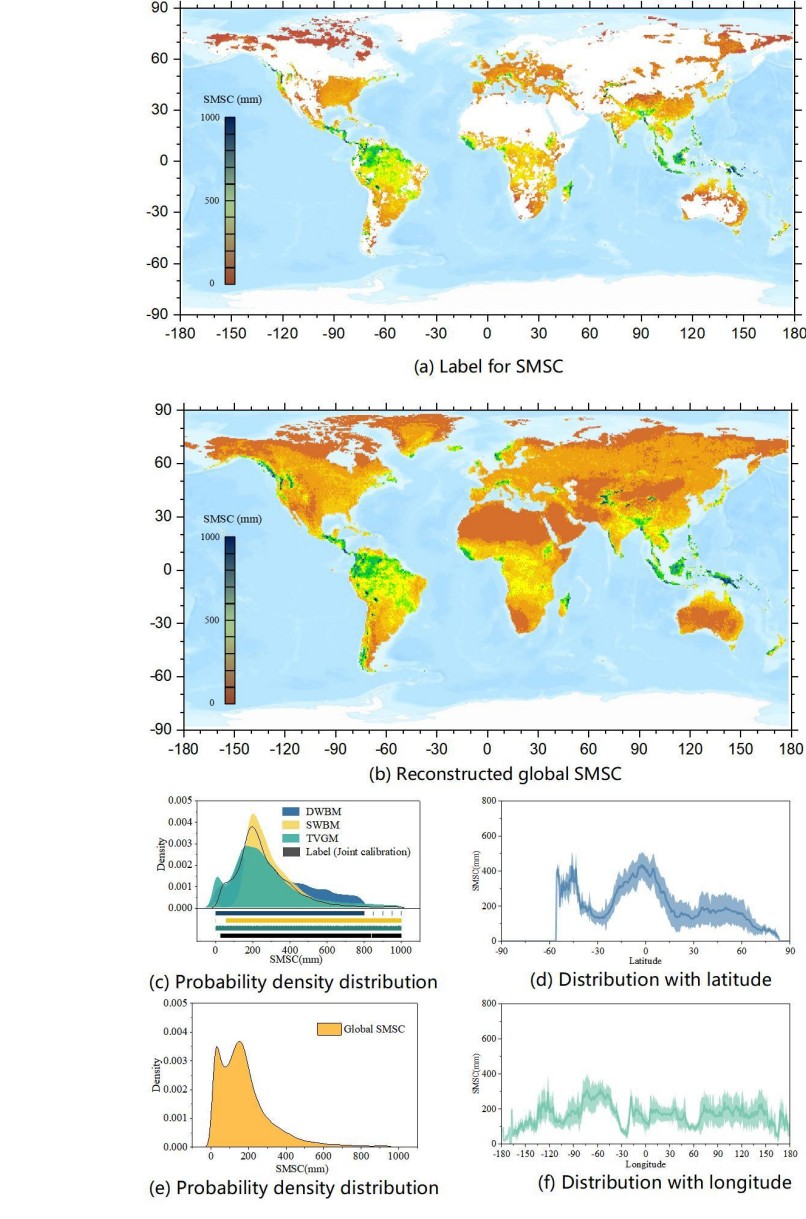


**Figure 2. Spatial distribution of labels and construction results for global soil moisture storage capacity (SMSC) parameters. (a) The spatial distribution of the label for deep learning of jointly calibrated SMSC values. (b) The spatial distribution of the constructed global SMSC parameters. (c) The probability density distribution of SMSC parameters calibrated in the labeling area. (e) The probability density distribution of constructed global SMSC parameters. (d) and (f) The distribution of variations of global SMSC with latitude and longitude.**



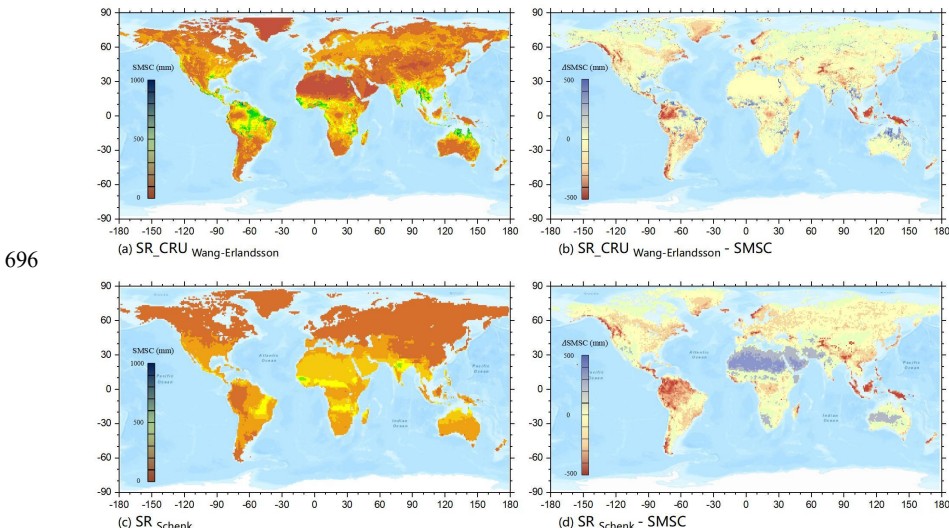

**Figure 3. Spatial distribution of other parameter datasets and the differences with global soil moisture**
**storage capacity (SMSC) parameters. (a) The spatial distribution of root zone storage capacity at 0.5°**
**resolution by Wang-Erlandsson et al. (2016). (c) The spatial distribution of rooting depth at 0.5° resolution**
**by Schenk et al. (2009). (b) and (d) The differences between global SMSC parameters and other parameter**
**datasets.**

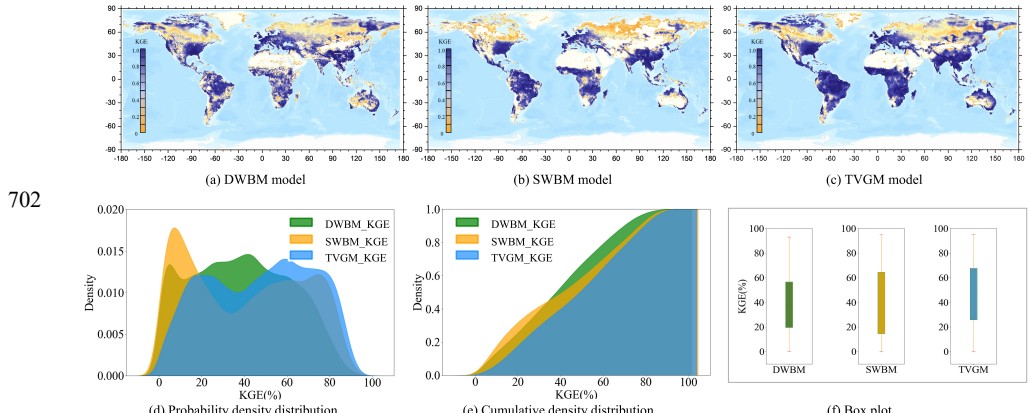


**Figure 4. Global distribution of Kling-Gupta efficiency (KGE) simulated by GRUN runoff depth on**
**grid-scale for three monthly water balance models. (a)The KGE of Dynamic Water Balance Model (DWBM).**
**(b) The KGE of Snowbelt-based Water Balance Model (SWBM). (c) The KGE of Time-variant Gain Model**
**(TVGM). (d) The probability density distributions of KGE for three models. (e) The cumulative density**
**distributions of KGE for three models. (f) The boxplot of KGE for three models.**


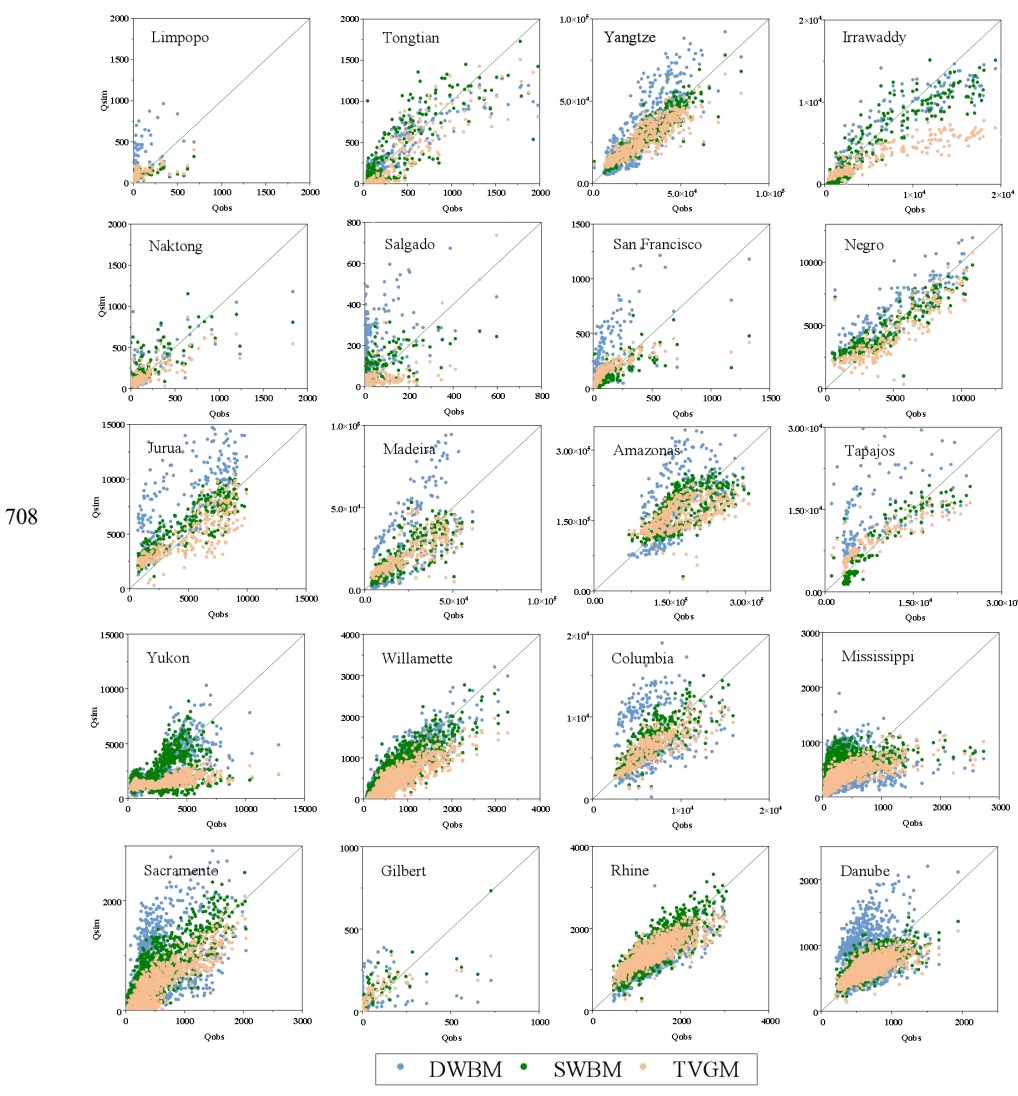


**Figure 5. Monthly observed streamflow versus simulated streamflow for three monthly water balance models in typical catchments. Blue dots represent Dynamic Water Balance Model (DWBM). Green dots represent Snowbelt-based Water Balance Model (SWBM). Orange dots represent Time-variant Gain Model (TVGM).**

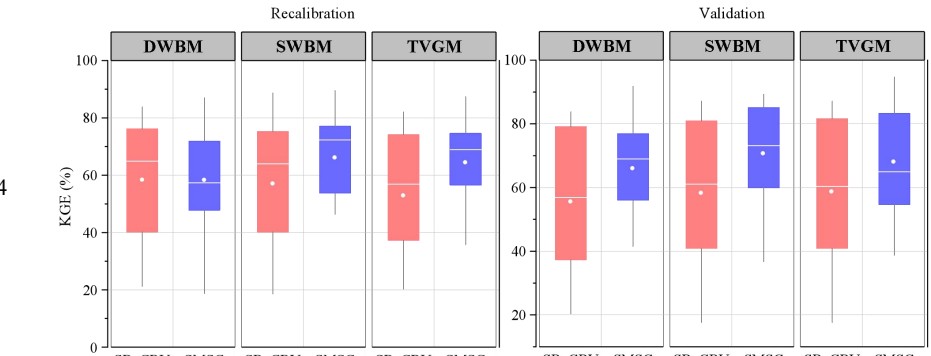

**Figure 6. Comparison of constructed SMSC parameter and root zone depth in runoff simulation for three monthly water balance models. The left figure represents the simulation accuracy during recalibration period. The right figure represents the simulation accuracy during validation period. The dots mean the average values, and the lines mean the median values.**


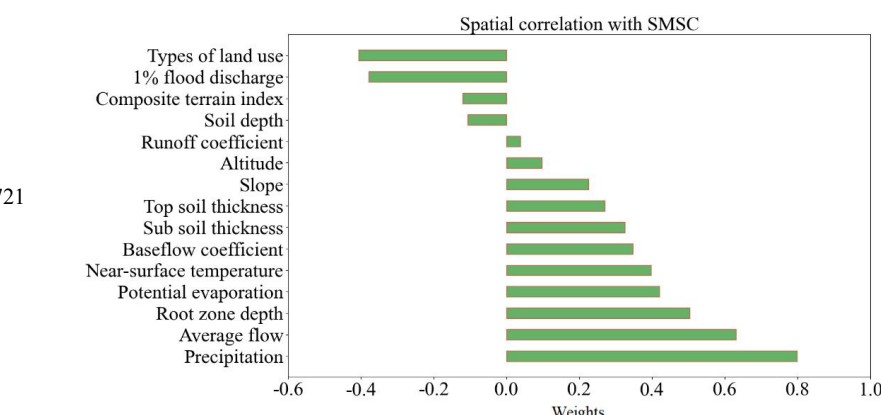

**Figure 7. Spatial correlation between SMSC and input variables by the permutation importance measure. Weights of permutation importance estimate represent the spatial correlation. The larger the absolute value means the more relevant variables.**

