# Peer review of "Global soil moisture storage capacity at 0.5° resolution for geoscientific modelling"

_Earth System Science Data, 2022_

## Referee Comment (RC2)

[referee-annotated manuscript omitted]

---

## Author Comment (AC1)

**Wuhan University**

**State Key Laboratory of Water Resources and Hydropower Engineering Science**

**Department of Hydrology and Water Resources**

Professor Pan Liu

Wuhan, China

Associate Editor, Journal of Water Resources Planning and Management, ASCE

Vice Director, State Key Laboratory of Water Resources and Hydropower Engineering Science

Head, Department of Hydrology and Water Resources

Tel: +8613871359778

Email: liupan@whu.edu.cn

**Date: October 13th, 2022**

**Dear Editor and Anonymous Referee #1,**

On behalf of my co-authors, we thank you for the constructive comments and suggestions, which significantly improved the manuscript (**NO. ESSD-2022-217**) entitled "Global soil moisture storage capacity at 0.5° resolution for geoscientific modelling".

The manuscript has been revised based on the comments from editors and reviewers. A point-by-point response to the reviews with referencing to the lines of the manuscript is attached to this letter. All the changes are marked in blue in the reply.

Thank you very much for handling our manuscript. We hope that you will find it to your satisfaction and we look forward to hearing from you in the near future.

Sincerely Yours

Pan Liu

On behalf of all co-authors

**Response to Reviewer #1**

This paper developed global soil moisture storage capacity (SMSC) map at $0.5° \times 0.5°$ grid scale, which provide a great improvement on the further application of hydrologic model in ungagged area. The new SMSC data was generated by the joint calibration of three hydrologic model and expand to global by deep learning networks, and was evaluated in 20 watersheds from 5 different climate regions. Overall, this manuscript is reasonably organized, and I think this manuscript is acceptable for publication with minor revision.

**Response:**

Thank you for providing the opportunity to revise further and submit the manuscript. We have studied the comments carefully and then edited the manuscript. In the following section, we summarize our responses to the comments. We believe that our responses have well addressed all concerns.

Specific comments

1. Line 27: "SMSC[L]" to "SMSC"

**Response:**

We appreciate your comments.

Soil moisture storage capacity **(SMSC)** is defined as the total amount of water stored in the soil within the plant root zone, one of the essential parameters linking the atmosphere and terrestrial ecosystems in the hydrological components (Chen, 2014; Mccormick et al., 2021).

Chen, B.: Analysis of hydrologic systems at multiple spatial scales and its implications for aggregating hydrologic process, Dissertations & Theses Gradworks, 2014.

McCormick, E. L., Dralle, D. N., Hahm, W. J., Tune, A. K., Schmidt, L. M., Chadwick, K. D., and Rempe, D. M.: Widespread woody plant use of water stored in bedrock, Nature, 597, 225-229, 10.1038/s41586-021-03761-3, 2021.

2. Line 99-100: According to Table 1, "1902 to 2014" means January 1902 to December 2014, hence there are 113 years in total. But in line 100, "first year···, 80 years···, 30 years···" only 111 years in all. Besides, does it enough to have only one year warming-up period? I suggest to have 3-5 years for warming-up.

**Response:**

Thank you for the valuable suggestion.

The data for the first **3 years** is used for warm-up, 80 years for calibration, and the remaining 30 years for validation.

3. Line 152: Please specify the calculation method of the Em.

**Response:**

Thank you. $E_m$ is the evaporation capacity of the basin, generally calculated from meteorological data (Wang et al., 2014). The $E_m$ is calculated by the Priestly-Taylor equation (Du Bruin et al, 1979). Meteorological data are required, including the temperature, the atmospheric pressure, the humidity, and the net radiation.

$$E_m = \alpha_e \frac{\Delta}{\Delta + \gamma}(R_n - G)$$

where $\alpha_e$ is the so-called Priestly-Taylor parameter which is taken as 1.26 if simplified, $\Delta$ is the slope of the saturation specific humidity-temperature curve, $\gamma$ is parameter of the heat of air at constant pressure, $R_n$ is the net radiation, and $G$ is the surface heat flux.

Wang, G., Zhang, J., Jin, J., Liu, Y., He, R., Bao, Z., Liu, C., and Li, Y.: Regional calibration of a water balance model for estimating stream flow in ungauged areas of the Yellow River Basin, Quaternary International, 336, 65-72, 10.1016/j.quaint.2013.08.051, 2014.

De Bruin H A R, Keijman J Q. The Priestley-Taylor evaporation model applied to a large, shallow lake in the Netherlands[J]. Journal of Applied Meteorology, 898-90310.1175/1520-0450(1979)018<0898:TPTEMA>2.0.CO;2. 1979.

4. Line 161: "SC" to "SMSC"

**Response:**

Thank you very much. This part is revised as follows:

where $S(t)$ is the soil humidity at the beginning of the period, $SMSC$ is the saturated soil humidity, $P$ is rainfall, $g_1$ and $g_2$ are the related parameters of the time-varying gain factor, where $g_1$ is the runoff coefficient after soil saturation, $g_2$ is the soil moisture influence coefficient.

5. Line 290: In figure 2(d), there is an increasing trend from -30° to -50° latitude. It's not decreasing towards the South Pole. Could you explain it?

**Response:**

Sorry for the confusion. This is mainly due to the limitations from the runoff data. Because there is no available measured runoff information in the south of 50°S mainly for the land of Antarctica and the ocean. Additionally, the GRUN grid runoff product is only available form 90°N to 50°S. The range of figure 2(d) has been modified as follows:

[Figure]

**Figure 2**. Spatial distribution of labels and construction results for global soil moisture storage capacity (SMSC) parameters. (d) The distribution of variations of global SMSC with latitude.

6. Line 334: "snowbelt" to "snowmelt"

**Response:**

Thank you. Sorry for this mistake. The sentence is revised as follows:

These three models do not take the temperature as the input, and therefore the **snowmelt** module is not considered.

7. Line 647: Table 4, could you add a column to list the climate zone of each catchment?

**Response:**

Thanks for your suggestions. A column is added to list the climate zone of each catchment as follows. The Köppen-Geiger climate clasification can be download in https://doi.org/10.1038/sdata.2018.214.

**Table 4. Validation of global SMSC parameters in typical catchments.**

| Number | Site name | Longitude | Latitude | Drainage area (km²) | Climate zone[1] | River | KGE (%) of DWBM model | | KGE (%) of SWBM model | | KGE (%) of TVGM model | | Basin average SMSC (mm) |
|---|---|---|---|---|---|---|---|---|---|---|---|---|---|
| | | | | | | | Cal[2] | Val[3] | Cal | Val | Cal | Val | |
| 1196551 | Beibrug | 29.99 | −22.23 | 201001 | Bsh | Limpopo | 47.06 | 74.15 | 53.25 | 69.67 | 43.11 | 41.67 | 149.84 |
| 2181500 | Zhimenda | 96.6 | 33.43 | 137704 | ET | Tongtian | 49.54 | 72.26 | 69.95 | 77.82 | 84.71 | 80.09 | 121.48 |
| 2181900 | Datong | 117.62 | 30.77 | 1705383 | Cfa | Yangtze | 55.87 | 84.52 | 86.48 | 91.88 | 85.75 | 91.16 | 206.85 |
| 2260500 | Sagaing | 96.1 | 21.98 | 117900 | Aw | Irrawaddy | 78.27 | 76.35 | 70.93 | 68.64 | 63.22 | 56.99 | 365.65 |
| 2694450 | Waegwan | 128.39 | 36 | 11195 | Dwa | Naktong | 67.81 | 58.89 | 66.63 | 41.78 | 77.99 | 44.93 | 231.37 |
| 3268270 | Caimancito | −64.47 | −23.73 | 25800 | Bsk | San Francisco | 67.54 | 78.11 | 53.66 | 83.31 | 63.44 | 63.55 | 228.89 |
| 3618090 | Cucui | −66.85 | 1.22 | 61781 | Af | Negro | 69.13 | 72.83 | 69.54 | 67.53 | 89.36 | 89.72 | 226.27 |
| 3624120 | Gaviao | −66.85 | −4.84 | 162000 | Af | Jurua | 49.13 | 66.07 | 71.06 | 69.88 | 88.35 | 80.24 | 532.84 |
| 3627030 | Manicore | −61.30 | −5.82 | 1126700 | Af | Madeira | 87.15 | 71.24 | 68.83 | 72.46 | 73.24 | 86.55 | 370.19 |
| 3629000 | Obidos-Porto | −55.51 | −1.95 | 4640300 | Af | Amazonas | 73.55 | 80.47 | 58.66 | 58.92 | 57.02 | 54.63 | 388.80 |
| 3629150 | Fortaleza | −57.64 | −6.05 | 358657 | Am | Tapajos | 39.03 | 49.10 | 87.55 | 74.90 | 75.24 | 63.62 | 428.36 |
| 3650745 | Ico | −38.87 | −6.41 | 12000 | Bsh | Salgado | 39.22 | 46.87 | 54.93 | 63.24 | 58.56 | 94.82 | 392.60 |
| 4103800 | Eagle AK | −141.20 | 64.79 | 293965 | Dfc | Yukon | 70.56 | 77.55 | 36.05 | 46.80 | 37.01 | 38.99 | 95.21 |
| 4115100 | Salem, OR | −123.04 | 44.94 | 18855 | Dsb | Willamette | 86.86 | 89.73 | 80.01 | 86.28 | 59.46 | 66.52 | 475.76 |
| 4115201 | Beaver, OR | −123.18 | 46.18 | 665371 | Dsb | Columbia | 58.60 | 47.52 | 79.14 | 76.35 | 88.81 | 74.00 | 358.43 |
| 4119100 | Paul, MN | −93.11 | 44.93 | 95312 | Cfa | Mississippi | 22.95 | 14.15 | 60.29 | 23.76 | 60.88 | 55.06 | 186.30 |
| 4146281 | Verona, CA | −121.60 | 38.77 | 55040 | Csa | Sacramento | 43.65 | 64.24 | 70.63 | 60.40 | 89.41 | 88.84 | 344.22 |
| 5109170 | Rockfields | 142.88 | −18.20 | 10987 | Bsh | Gilbert | 52.02 | 76.95 | 13.20 | 50.34 | 73.34 | 52.15 | 245.60 |
| 6335180 | Worms | 8.38 | 49.64 | 68827 | Cfb | Rhine | 73.97 | 76.66 | 78.43 | 84.00 | 76.88 | 78.37 | 296.07 |
| 6342800 | Hofkirchen | 13.12 | 48.68 | 47496 | Cfb | Danube | 56.53 | 46.58 | 61.31 | 53.67 | 69.49 | 61.30 | 247.41 |
| | | | | | | Mean KGE | 59.42 | 66.21 | 64.53 | 66.08 | 70.76 | 68.16 | —— |

[1] Köppen-Geiger climate clasification
[2] Calibration period
[3] Validation period

Beck, H.E., N.E. Zimmermann, T.R. McVicar, N. Vergopolan, A. Berg, E.F. Wood: Present and future Köppen-Geiger climate classification maps at 1-km resolution, Scientific Data 5:180214, doi:10.1038/sdata.2018.214 (2018).

---

## Author Comment (AC2)

**Wuhan University**

State Key Laboratory of Water Resources and Hydropower Engineering Science Department of Hydrology and Water Resources

Professor Pan Liu Wuhan, China Associate Editor, Journal of Water Resources Planning and Management, ASCE Vice Director, State Key Laboratory of Water Resources and Hydropower Engineering Science Head, Department of Hydrology and Water Resources Tel: +8613871359778 Email: liupan@whu.edu.cn

**Date: October 26th, 2022**

**Dear Editor and Anonymous Referee #2,**

On behalf of my co-authors, we thank you for the constructive comments and suggestions, which significantly improved the manuscript (**NO. ESSD-2022-217**) entitled "Global soil moisture storage capacity at 0.5° resolution for geoscientific modelling".

The manuscript has been revised based on the comments from editors and reviewers. A point-by-point response to the reviews with referencing to the lines of the manuscript is attached to this letter. All the changes are marked in blue in the reply.

Thank you very much for handling our manuscript. We hope that you will find it to your satisfaction and we look forward to hearing from you in the near future.

Sincerely Yours

anlie

Pan Liu On behalf of all co-authors

**Response to Reviewer #2**

**Specific comments**

1. Line 99: There are 113 years in total. However, the authors only indicated 111 years. How about the other two years? How do you select the years for calibration and for validation?

**Response:**

Thank you. The other two years are used for warm-up. The warm-up is an adjustment process for the model to reach an optimal state. The sentences have been revised as follows:

Monthly measurements cover the year from 1902 to 2014 in the global grids. The data for the **first 3 years** is used for warm-up, 80 years for calibration, and the remaining 30 years for validation.

2. Line 319: What is CCN?

**Response:**

Sorry for the mistake. CNN represents convolutional neural networks. The sentence has been revised as follows:

The model parameter SMSC among the three monthly water balance models is the same in the proposed dataset constructed by CNN.

3. Line 391: "Secondly, there are also errors in the measurement data of the observation station, which leads to the uncertainty of the input data." What kind of errors?

**Response:**

We appreciate your comments. The sentence has been revised as follows:

Secondly, the errors also come from the observation uncertainty of the input data. Every stage of hydrological modelling acquires some uncertainty. This uncertainty can be broadly grouped into input forecast uncertainty and hydrological model uncertainty. The input forecast uncertainty due to input data such as precipitation, temperature and other metrological inputs to the model (Singh and Dutta, 2017).

Singh, S.K. and Dutta, S., 2017. Observational uncertainty in hydrological modelling using data depth. Glob. Nest J, 19: 489-497.

4. Line 641: Table 1 Please introduce the accuracy level of every data used in this study.

**Response:**

Thank you very much.

The Climatic Research Unit Timeseries dataset (CRU TS) is a widely used climate dataset on a  $0.5^{\circ}$  latitude by  $0.5^{\circ}$  longitude grid over all land domains of the world except Antarctica. It is derived by the interpolation of monthly climate anomalies from extensive networks of weather station observations. CRU TS has good high-frequency agreement with CRUTEM4.6 (correlation coefficient, R = 0.99 globally), UDEL (R = 0.97 globally) and JRA-55 (R =0.99 globally, 1958-2017 only) (Harris et al., 2020).

The Global Streamflow Characteristics Dataset (GSCD) consists of global maps of 17 streamflow characteristics, providing information about runoff behavior for the entire land surface including ungauged regions. It was constructed by streamflow observations from a highly heterogeneous set of 3394 catchments (<10,000 km2) worldwide. The maps were compared to equivalent maps derived from the simulated daily runoff of four macroscale hydrological models (Beck et al., 2015).

Soil and vegetation dataset is provided by high-resolution estimates within a global 30 arc-second (~1 km) grid using the best available data for topography, climate, and geology as input (Pelletier et al., 2016).

Topography dataset contains elevation-based parameters at  $0.5^{\circ}$  spatial resolutions that were developed to support a wide variety of global modeling activities. It is the highest resolution database of global coverage of standard elevation-based derivatives (Verdin et al., 2011).

The Global Runoff Data Centre (GRDC) is built to provide a global observed hydrological data set to complement a specific set of atmospheric data in the framework of the First Global GARP Experiment (FGGE). Today the database comprises discharge data of well over 10,000 gauging stations from all over the world .

The global grid runoff depth database (GRUN) dataset contains a gridded global reconstruction of monthly runoff timeseries. On average GRUN shows higher predictive skills than a collection of the global hydrological models, especially with respect to the reproduction of the seasonality, dynamics and anomalies of runoff (Ghiggi et al., 2019).

Therefore, the spatial and temporal resolution of the input information has been added in Table 1 as follows:

| Data type                | Data                                                                                      | Spatial    | Temporal               | Data/product sources                                                                                                                                                                 |
|--------------------------|-------------------------------------------------------------------------------------------|------------|------------------------|--------------------------------------------------------------------------------------------------------------------------------------------------------------------------------------|
| Meteorology              | Precipitation
Potential
evaporation
Near-surface
temperature                  | 0.5 degree | 1901 - 2018
Monthly | Cru TS 4.03 monthly high-resolution grid data
https://data.ceda.ac.uk/badc/cru/data/cru_ts/cru_ts_4.
03                                                                        |
| Soil and
vegetation   | Soil thickness                                                                            | 0.5 degree | Static                 | Global 1km grid soil, weathering layer, and
sedimentary layer thickness published by
ORNL in 2016
https://daac.ornl.gov/SOILS/guides/Global_Soil_Reg
olith_Sediment.html |
|                          | Root zone depth
Soil type                                                              |            |                        | Stockholm university in 2016
http://dx.doi.org/10.5194/hess-20-1459-2016-
supplement                                                                                           |
|                          |                                                                                           |            |                        | Upper and lower global soil type data released
by USDA
https://www.nrcs.usda.gov/wps/portal/nrcs/detail/soil
s/use/                                                         |
|                          | Types of land use                                                                         |            |                        | AVHRR land-use types issued by NOAA                                                                                                                                                  |
| Topography               | Slope
Altitude
Composite terrain
index CTI                                       | 0.5 degree | Static                 | GMT global 0.5-degree terrain data released by
ISLSCP in 2010
https://daac.ornl.gov/cgi-bin/dsviewer.pl?ds_id=1007                                                             |
| Runoff
characteristic | Average flow
Runoff
coefficient
Baseflow
coefficient
1% flood
discharge | 0.5 degree | Static                 | GSCD global runoff data set released
by GloH2O in 2015
http://www.gloh2o.org/gscd/                                                                                             |
| Runoff                   | Runoff of
catchment
stations                                                        | Stations   | 1991 - 2010
Monthly | GRDC global runoff data center
(Including data from more than 10000 stations
around the world)
https://www.bafg.de/GRDC/EN/Home/homepage_no
de.html                      |
|                          | Grid runoff                                                                               | 0.5 degree | 1902 - 2014
Monthly | GRUN global grid runoff depth database
released by the Federal Institute of technology
https://doi.org/10.6084/m9.figshare.9228176                                             |

**Table 1. Research data and sources.**

Harris, I., Osborn, T.J., Jones, P. and Lister, D., 2020. Version 4 of the CRU TS monthly high-resolution gridded multivariate climate dataset. Scientific Data, 7(1): 1-18.

Beck, H.E., de Roo, A. and van Dijk, A.I.J.M., 2015. Global Maps of Streamflow Characteristics Based on Observations from Several Thousand Catchments\*. Journal of Hydrometeorology, 16(4): 1478-1501.

Pelletier, J.D. et al., 2016. A gridded global data set of soil, intact regolith, and sedimentary deposit thicknesses for regional and global land surface modeling. Journal of Advances in Modeling Earth Systems, 8(1): 41-65.

Verdin, K.L. et al., 2011. ISLSCP II HYDRO1k elevation-derived products.

Ghiggi, G., Humphrey, V., Seneviratne, S.I. and Gudmundsson, L., 2019. GRUN: an observation-based global gridded runoff dataset from 1902 to 2014. Earth System Science Data, 11(4): 1655-1674.

5. Line 645: Table 3 Why do you only test the performance of image window at  $5 \times 5$  and  $10 \times 10$ ?

**Response:**

Thank you for the comments.

The selection of the image window is a trade-off between the level of accuracy and the speed of computing. The image window at  $5\times5$  corresponds to the influence of a  $2.5^{\circ}$  square (approximately 250 km) on the center point, while  $10\times10$  corresponds to the influence of a  $5^{\circ}$  square (approximately 500 km). It is generally within this distance that the spatial variables are similar, and it is meaningless in farther distance. Additionally, we added the results for the smaller image window( $3\times3$ ). The results show that the recognition network is poor if the image window is too small. The computational burden from the increase in image windows is no longer as cost-effective as the increase in the correlation coefficient for the image window from  $5\times5$  to  $10\times10$ .

| Image window | Time interval | Loss function | Evaluating indicator |        |
|--------------|---------------|---------------|----------------------|--------|
| image window | Time interval | MSE           | MAE                  | $R^2$  |
| 2×2          | Training set  | 0.0116        | 0.0752               | 0.7256 |
| 3×3          | Test set      | 0.0252        | 0.0964               | 0.4867 |
| EXE          | Training set  | 0.0032        | 0.0411               | 0.8932 |
| 3×3          | Test set      | 0.0170        | 0.0666               | 0.5935 |
| 10×10        | Training set  | 0.0021        | 0.0341               | 0.9139 |
| 10×10        | Test set      | 0.0061        | 0.0510               | 0.7597 |

Table 3. Results of construction model of global soil moisture storage capacity (SMSC) parameters.

6. Line 695: Figure 2 The legends of (a) and (b) do not match the color of the content.

**Response:**

Thank you. The legends of (a) and (b) have been modified as follows: